# *Zizania latifolia* and Its Major Compound Tricin Regulate Immune Responses in OVA-Treated Mice

**DOI:** 10.3390/molecules27133978

**Published:** 2022-06-21

**Authors:** Jae-Yeul Lee, Se-Ho Park, Kwang-Hwan Jhee, Seun-Ah Yang

**Affiliations:** 1Institute of Natural Science, Keimyung University, Daegu 42601, Korea; sunriseow@naver.com (J.-Y.L.); p86ks1@naver.com (S.-H.P.); 2Department of Applied Chemistry, Kumoh National Institute of Technology, Gumi 39177, Korea; khjhee@kumoh.ac.kr; 3Department of Food Science and Technology, Keimyung University, Daegu 42601, Korea

**Keywords:** flavonoids, tricin, *Zizania latifolia* extract, anti-allergic activity, immunomodulation

## Abstract

Tricin, a flavone belonging to the *Gramineae* family, has been confirmed to be the primary compound in a *Zizania latifolia* extract (ZLE) that prevents allergies. Various allergic reactions occur because of the unbalanced differentiation of T help cells (Th) and the consequent overproduction of IgE. Therefore, the regulation of Th1 and Th2 responses by T helper cell differentiation is essential for suppressing allergic responses. This study confirmed the immunomodulatory effects of ZLE and the major compound tricin in an OVA-sensitized mouse model. The IgE and OVA-specific production of tricin and ZLE in plasma were investigated in OVA-sensitized mice. The effects of tricin and ZLE on the amount of Th1 and Th2 cytokines and transcription factors released in splenocytes were investigated in OVA-sensitized mice. The skin roughness and the number of mast cells were confirmed by staining the skin surface with H&E and toluidine blue. Tricin and ZLE reduced the plasma IgE and OVA-specific-IgE levels significantly compared to the OVA group. On the other hand, tricin and ZLE promoted the release of the Th1 cytokines IL-12 and IFN-γ and inhibited the release of Th2 cytokines (IL-4, -10, -13, and -5) in OVA-sensitized mice. Tricin and ZLE induced T-bet and NFATc2 expression, and-down regulated GATA-3 levels. The skin roughness and the number of mast cells decreased in the OVA-immunized mice. Overall, the data indicate that tricin and ZLE may prevent allergy-related diseases through immunomodulation.

## 1. Introduction

Mast cells are essential for acute and chronic inflammation and immediate allergic hypersensitivity reactions and also play a significant role in immunoglobulin-E (IgE)-mediated allergic responses. IgE and specific allergens bind to FcεRI, which is an IgE receptor that is expressed on the surface of mast cells. As the intracytoplasmic calcium concentration and degranulation increase, histamine and inflammatory cytokines are secreted to dilate blood vessels and increase vascular permeability to cause allergic reactions. The pathological phenomenon of allergies was studied based on the Th2 hypothesis in relation to the immunological mechanism. The incidence of allergies was explained by focusing on the Th1/Th2 balance. Th1 cells that secrete IL-2 and IFN-γ and Th2 cells that secrete IL-4 and IL-5 exist in T help cells, which cause an immune response in mice. Hence, the Th1 and Th2 responses are antagonistic and maintain a balance. Moreover, an imbalance in the Th1/Th2 responses is a key factor in the immunological etiology of allergies. Furthermore, Th1 and Th2 cell differentiation and cytokines production (IL-4, -6, -13, TNF-α) in these cells can be controlled using various cellular transcription factors (GATA-3, T-bet) [1,2,3]. According to Yates et al., Cañas et al., and Gandihi et al., Th1 and Th2 responses have been applied to in vivo studies, and Th1/Th2 reagents and extracts have been developed [4,5,6]. Anti-inflammatory drugs, antihistamines, or anti-leukotriene drugs are prescribed and used as therapeutic agents for inflammatory and allergic diseases. In addition, recent studies have been performed on anti-allergy compounds from natural substances that effectively prevent skin aging and inflammatory diseases, such as atopic dermatitis, due to the increasing interest in healthy aging due to the increase in the aging population [7,8,9].

*Zizania latifola* (*Gramineae*) is a wild rice plant from Asia that belongs to the *Zizania* genus. *Z. latifolia* grains have strong biological effects, such as hyperlipidemia suppression [10], the ability to eliminate oxidative stress from cells [11], blood glucose level suppression [12], insulin resistance improvement [12], and anti-obesity [10]. In addition, anti-inflammatory compounds derived from tricin [13] that have been isolated as active ingredients from stem extracts have been reported. Additionally, Park et al. [14] and Moon et al. [15] reported the effects of tricin and *Z. latifolia* extract’s anti-wrinkle activity both in vitro and in vivo. Nevertheless, despite the high efficacy of *Z. latifolia*, systematic research on its use as a functional material has been insufficient. In our previous study, we identified five tricin derivatives (Figure 1) in methanol extract using the aerial part of *Z. latifolia* [13]. Lee et al. [13] reported that the tricin derivatives prevented the release of β-hexosaminidase in an anti-IgE-stimulated allergic reaction in RBL-2H3 mast cells, and the tricin content in *Z. latifolia* extract was calculated by means of LC-MS/MS analysis (0.1% tricin in *Z. latifolia* extract). According to the reports of Mohanlal et al. [16], Kwon and Kim [17] and Quan et al. [18], the tricin content of *Oryza sativa* calculated 48.6 mg/kg, 32.9 mg/kg, and 0.55 mg/kg, respectively. Tricin suppressed the arachidonic acid signal, mitogen-activated protein kinase (MAPK), and FcεRI signal of the RBL-2H3 cells and showed anti-allergic effects by suppressing the release of LTB4, LTC4, and PGE2 [19]. Indeed, Jiang et al. [20] revealed that tricin has effectively the biological properties by functional groups such as phenolic hydroxyl groups, carbonyl groups, methoxyl groups. In this study, a mouse allergy model (ovalbumin (OVA)-sensitized BALB/c mice) was used to examine the anti-allergic activities of ZLE or tricin.

## 2. Results and Discussions

### 2.1. Body Weight, ALT, and AST Analysis in ZLE or Tricin Administration

No changes were observed for body weight, regardless of whether ZLE or tricin was administered (Figure 2A). The effects of tricin and ZLE on the AST and ALT levels in blood were also investigated to confirm that there were no hepatotoxic effects. Both the AST and ALT were increased in the OVA group (Figure 2B,C). On the other hand, there was an increase in the reference range [21,22,23,24]. Neither ZLE (low-dose group and high-dose group) nor tricin (low-dose group and high-dose group) altered the ALT and AST contents in the serum compared to the OVA group. These results suggest that the administration of ZLE and tricin does not affect hepatotoxicity.

### 2.2. Effects of ZLE or Tricin on the Release of IgE or OVA-Specific IgE

In a mouse model, OVA treatment induces IgE production and OVA-specific IgE [25]. IgE binds to FcεRI and FcεRII of mast cells and cross-links with the antigen to cause degranulation. Various inflammatory mediators, such as leukotrienes, histamines, and prostaglandins, are secreted because of degranulation. These mediators are involved in allergic reactions [26]. The Th2 and IgE response should be amplified. Thus, this study, we investigated the effects of ZLE or tricin on of IgE release and OVA-specific IgE by ELISA. The OVA stimulation in mice increased by 1440 pg/mL and 14.61 pg/mL compared to the control group, respectively (Figure 3A,B). The ZLE or tricin groups showed a decrease in the IgE or OVA-specific IgE compared with the OVA-treated group (ZLE: 603.9 pg/mL and 5.18 pg/mL at a 300 mg/kg dosing level; tricin: 803.6 pg/mL and 8.2 pg/mL at a 300 µg/kg dosing level), indicating that ZLE and tricin have strong anti-allergic activity and can reduce inflammation. The dexamethasone group inhibited the IgE or OVA-specific IgE at approximately 478 pg/mL and 5.0 pg/mL, respectively, indicating that dexamethasone inhibits B cells reaction. OVA-sensitive mice are often used to evaluate anti-allergic activity. Some oral substances have been shown to regulate allergic reactions and inhibit the total IgE or the levels of OVA-specific IgE in the serum of OVA-sensitive mice [27,28].

In the present study, the oral administration of tricin and ZLE in OVA-treated mice suppressed IgE and OVA-specific IgE in serum, similar to the reduction in dexamethasone-treated mice. These results suggest that the inhibition of the IgE and OVA-specific IgE reduces the amount of IgE binding to the FcεRI in mast cells, thereby reducing allergic reactions [29].

### 2.3. Effects of ZLE or Tricin on the Secretion of Th1 and Th2 Cytokines

An increase in the Th2 immune response in the immune system is characteristic of various allergic diseases. Th2 cytokines (IL-4, -5, -10, -13) activate the inflammatory pathways in various cell types, increase the IgE level, and decrease the Th1 immune response.

The secretions of inflammation cytokines, including IFN-γ, IL-12, -4, -10, -13, and -5 were measured in the OVA-treated mice to check the immuno-modulation effects of the ZLE and tricin. There was secretion of IFN-γ, -4, -10, -13, and -5 but not IL-12 in the OVA-stimulated group compared with the control group (Table 1). The production of Th1 cytokines (IL-12, and IFN-γ) increased in a dose-dependent manner in the groups receiving ZLE (150 and 300 mg/kg dosing level) and tricin (150 and 300 µg/kg dosing level) compared to the OVA-stimulated group. On the other hand, the OVA-induced Th2 cytokine levels (IL-4, -5, -10, and -13) were reduced significantly by the administration of the ZLE or tricin treatment. These results suggest that ZLE and tricin regulate Th1/Th2 balance selectively by increasing the Th1 cytokine levels and decreasing the Th2 cytokine levels.

The alleviation of the allergy symptoms has been shown to reduce serum IgE levels by modulating the Th1/Th2 balance from a Th2- to a Th1- dominant state [30].

In this study, 0.1% tricin was used for each ZLE concentration to confirm the efficacy of tricin on ZLE. In our previous report, the content of tricin in the ZLE was quantitatively analyzed and its contribution to the anti-allergy effect of the plant extract was evaluated in vitro [20]. In the present in vitro data, we calculated that the anti-allergic activity (IL-4) of tricin was approximately 55.60% compared to the ZLE, suggesting that the tricin in the plant strongly contributes to the anti-allergic activity. In the in vivo data, the anti-allergic activity (IL-4) of tricin was approximately 61.01% compared to the ZLE, suggesting that the tricin showed similar activity to the in vitro data.

### 2.4. Effects of ZLE or Tricin on the Balance of Th1 and Th2 Mediators

The effects of ZLE and tricin on the expression of T-bet, NFATc2, and GATA-3 were then examined by PCR and Western blot analysis. The OVA group showed increased levels of NFATc2 and GATA-3, but not T-bet (Figure 4A). The ZLE and tricin group showed increased T-bet and NFATc2 levels compared to the OVA group (2.65 and 2.26-fold at 300 mg/kg in the ZLE group and 300 μg/kg in the tricin group, 1.37 and 1.49-fold at 300 mg/kg in the ZLE group and 300 μg/kg in the tricin group, respectively). The ZLE and tricin groups suppressed the GATA-3 level (0.26 and 0.32-fold at 300 mg/kg in the ZLE group and 300 μg/kg in the tricin group, respectively).

In addition, the ZLE and tricin groups had increased T-bet and NFATc2 protein expression levels compared to the OVA group (2.03 and 1.79-fold at 300 mg/kg in the ZLE group and 300 μg/kg in the tricin group; 1.50 and 1.28-fold at 300 mg/kg in the ZLE group and 300 μg/kg in the tricin group, respectively) (Figure 4B). The ZLE and tricin groups also had suppressed GATA-3 protein expression (0.31 and 0.28-fold at 300 mg/kg in the ZLE group and 300 μg/kg in the tricin group, respectively). T-bet regulates Th1 cytokines, such as IFN-γ and IL-12 as well as GATA-3, which strongly transcribes the expression of Th2 cytokines (IL-4 and -5).

### 2.5. Histopathological Changes in ZLE or Tricin Administration in Mice

In mice, OVA-induced dermatitis resulted in destroyed skin tissue caused by the dermal infiltration of inflammatory cells [31]. The OVA group demonstrated higher dermal and epidermal thicknesses than the control group (Figure 5). Sensitization with OVA elicited a local cutaneous inflammatory response. The total number of mast cells was significantly higher in the OVA-sensitized mice than in the control group. This increase in OVA-induced mast cells was inhibited by each dose of dexamethasone, ZLE, and tricin. Toluidine Blue staining showed that the total number of mast cells decreased markedly, a finding that is consistent with the dramatic decrease in the allergic lesions of ZLE and tricin (Figure 6).

## 3. Materials and Methods

### 3.1. Reagents

MTT (3-(4,5-dimethylthiazol-2-yl)2-,5-diphenyltetrazolium bromide) was procured from Sigma Aldrich Co. (St. Louis, MO, USA). T-bet, GATA-3, and NFATc2 were purchased from Cell Signaling Technology (Beverly, MA, USA). ELISA kits for IL-4, -5, -10, -12, -13, and IFN-γ were supplied by e-Bioscience, Inc. (R&D Systems, Minneapolis, MN, USA). The ELISA kit used for the IgE OVA-specific IgE were supplied by Cyman Chemical Co. (Ann Arbor, MI, USA). Roswell Park Memorial Institute medium (RPMI 1640) and fetal bovine serum (FBS) were obtained from the American Type Culture Collection (Manassas, VA, USA). Tricin (4′,5,7-trihydroxy-3′,5′-dimethoxyflavone) was purchased from ChromaDex. Inc. (Los Angeles, CA, USA). All of the reagents used in this study were of analytical grade (purity: >99.9%).

### 3.2. Plant Material and Preparation of ZLE

ZLE was prepared by modifying the process that was previously reported by Moon et al. and Lee et al. [13,14,15,20]. *Z. latifolia* cultivated in Yeongcheon-si (Gyeongsangbuk-do, Korea) and *Z. latifolia* were purchased from the Pureunsan Agricultural Association Corporation (Dongdaemun-gu, Seoul, Korea). Dried leaves of *Z. latifolia* underwent extraction with 10 times its weight in 70% ethanol (Duksan Science, Seoul, Korea) at 80 °C for 6 h. The extracted solution was concentrated and lyophilized to produce ZLE. The lyophilized ZLE confirmed to have a yield of 15%.

### 3.3. Animal

Male BALB/c mice that were six weeks in age were sourced from Hanabio, Co Ltd. (Gyronggi, Korea). All of the mice were kept in an air-conditioned room and were allowed to acclimate for at one week. The animal protocol used in this study was reviewed and approved by the Southeast Medi-chem Institute Institutional Animal Care and Use Committee (SEMI-20-016).

### 3.4. The Immunoglobulin Levels and Cytokine Production

Figure 7 presents the in vivo design. The mouse allergic model was prepared according to methods that have been previously described [20,32]. Briefly, each mouse, with the exception of the control mice, were individually treated with OVA (20 µg, i.p.) and aluminum hydroxide (2.25 mg) on days 7 and 21. Then, the OVA-sensitized mice (*n* = 5) received orally administered ZLE (150, 300 mg/kg), tricin (150, 300 µg/kg), or dexamethasone (DEX; 0.5 mg/kg) on each day. The control mice were orally administrated distilled water (Table 2). On day 32, blood was collected, and plasma samples were stored at −80 °C. The total or specific-IgE levels in the plasma samples were measured and calculated using a mouse IgE detection kit (Cyman Chemical Co., Ann Arbor, MI, USA). To measure cytokine production, splenocytes were isolated from the OVA-treated, OVA + ZLE, and OVA + tricin groups (*n* = 5) on 32 day and were cultured with RPMI-1640 containing 10% FBS. Then, the isolated splenocytes were seeded (5 × 10^6^ cells/mL per well) into 24-well cell culture plates and incubated for three days (negative control: non-treated OVA, positive control: OVA at 100 mg/mL). The Th1 and Th2 cytokine (IL-4, -5, -10, -12, -13, IFN-γ) levels in the supernatants were measured and calculated using IL-4, -5, -10, -12, -13, and IFN-γ ELISA kits (R&D Systems, Minneapolis, MN, USA).

### 3.5. Reverse Transcriptase PCR (RT-PCR)

To check the gene levels of T-bet, NFATc2, and GATA-3, RT-PCR analyses were performed by modifying the methods of Park et al. [32] and Lee et al. [33]. The total RNA in the splenocytes was isolated using a TRIzol Reagent (GIBCO BRL, Carlsbad, CA, USA). cDNA was synthesized using an iScript cDNA synthesis kit (BioRad, Seoul, Korea). The levels of each of the target genes (T-bet, NFATc2, GATA-3) were normalized to the GAPDH level. The forward and reverse primers of T-bet, NFATc2, GATA-3, and GAPDH are described below Table 3. Each cDNA sample was amplified using the process below. Step 1, 5 min at 95 °C; step 2: 30 s at 95 °C; step 3: 45 s at 60 °C; step 4: 1 min at 72 °C; and step 5: 15 min at 72 °C. Each of the targeted bands (GATA-3, T-bet, NFATc2, GAPDH) were analyzed using agarose gel electrophoresis. The band intensity in the electrophoresed gels was measured using Image software (UVP Vision Works^®^ LS Image Acquisition & Analysis Software, Upland, CA, USA).

### 3.6. Immunoblotting Analysis

Immunoblotting analysis was performed following a previously described method [20]. Then, we investigated the protein samples prepared from splenocytes and antibodies specific to mouse T-bet, NFATc2, GATA-3, or β-actin by performing Western blot analysis. Splenocytes (1 × 10^7^ cells per group) were lysed in a lysis buffer (RIPA) containing protease inhibitor (P3100_001, GenDEPOT, Katy, TX, USA) and underwent freezing incubation for 60 min. The protein concentrations in each sample were measured using a BCA protein assay (Thermo Fisher Scientific, Waltham, MA, USA). The proteins (20 μg) were loaded with 10% sodium dodecyl sulfate (SDS)-polyacrylamide gel, and the separated proteins were transferred to a PVDF membrane (Whatman GmbH, Dassel, Germany). The transferred PVDF membrane was blocked with 5% skimmed milk in Tris-buffered saline (TBS) containing 0.1% Tween-20 for 1 h. Then, the blocked PVDF membrane was incubated with the primary antibodies (T-bet, NFATc2, GATA-3, and β-actin) for 16 h at 4°C. After the incubation of the primary antibodies, the PVDF membrane was washed three times with TBS containing 0.1% Tween-20, and the membrane was then incubated with horseradish peroxidase (HRP)-linked secondary antibodies (T-bet, NFATc2, GATA-3; anti-rabbit secondary antibody, β-actin; and anti-mouse secondary antibody) (Cell Signaling Technology, Beverly, MA, USA) at room temperature for 1 h. The targeted proteins (T-bet, NFATc2, GATA-3, β-actin) were measured using Image software (UVP Vision Works^®^ LS Image Acquisition & Analysis Software, Upland, CA, USA).

### 3.7. Histological Analysis

To investigate the histological changes and mast cell activation, H&E staining and toluidine blue staining were performed according to the method of Wang et al. [31] with slight modifications. Briefly, Skin specimens were fixed in 10% formalin (prepared in 0.1 M PBS), and 5 μm thick slide sections were prepared. The sections were then stained with hematoxylin and eosin (H&E) or with toluidine blue to examine the epidermal thickness and infiltration of inflammatory cells in the dermal area. The images of the stained sections were obtained at 100×. The thickness was measured in five randomly selected fields from each sample.

### 3.8. Statistical Analysis

Data are presented as the means ± SD values of three experiments. Each group was with Student’s *t*-test. Significant differences between each group are represented with *p* values (* *p* < 0.05, ** *p* < 0.01, and *** *p* < 0.001).

## 4. Conclusions

In conclusion, ZLE exerts anti-allergic effects in OVA-treated mice, and tricin is responsible for its prophylactic action against allergic diseases. The suppression mechanism on tricin that is involved in allergic reaction signaling was investigated via the GATA-3/T-bet signaling pathway, which is related to the secretions of Th1/Th2 cytokines (Th1 cytokines; IFN-γ, IL-5, Th2 cytokine; IL-4) in OVA-treated mice. In terms of biological activity, tricin regulates the Th1 and Th2 balance by selectively reducing Th2 cytokine level and increasing Th1 cytokine levels.

Overall, the data confirmed that tricin and ZLE have potential anti-allergic activity and that tricin can be used as a phytochemical to modulate immune cells in the development of functional products related to allergic disorders.

## Figures and Tables

**Figure 1 molecules-27-03978-f001:**
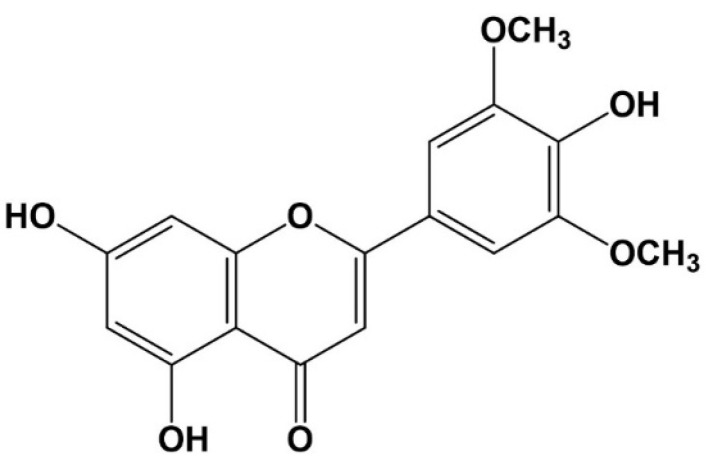
Tricin (4′,5,7-trihydroxy-3′,5′-dimethoxyflavone).

**Figure 2 molecules-27-03978-f002:**
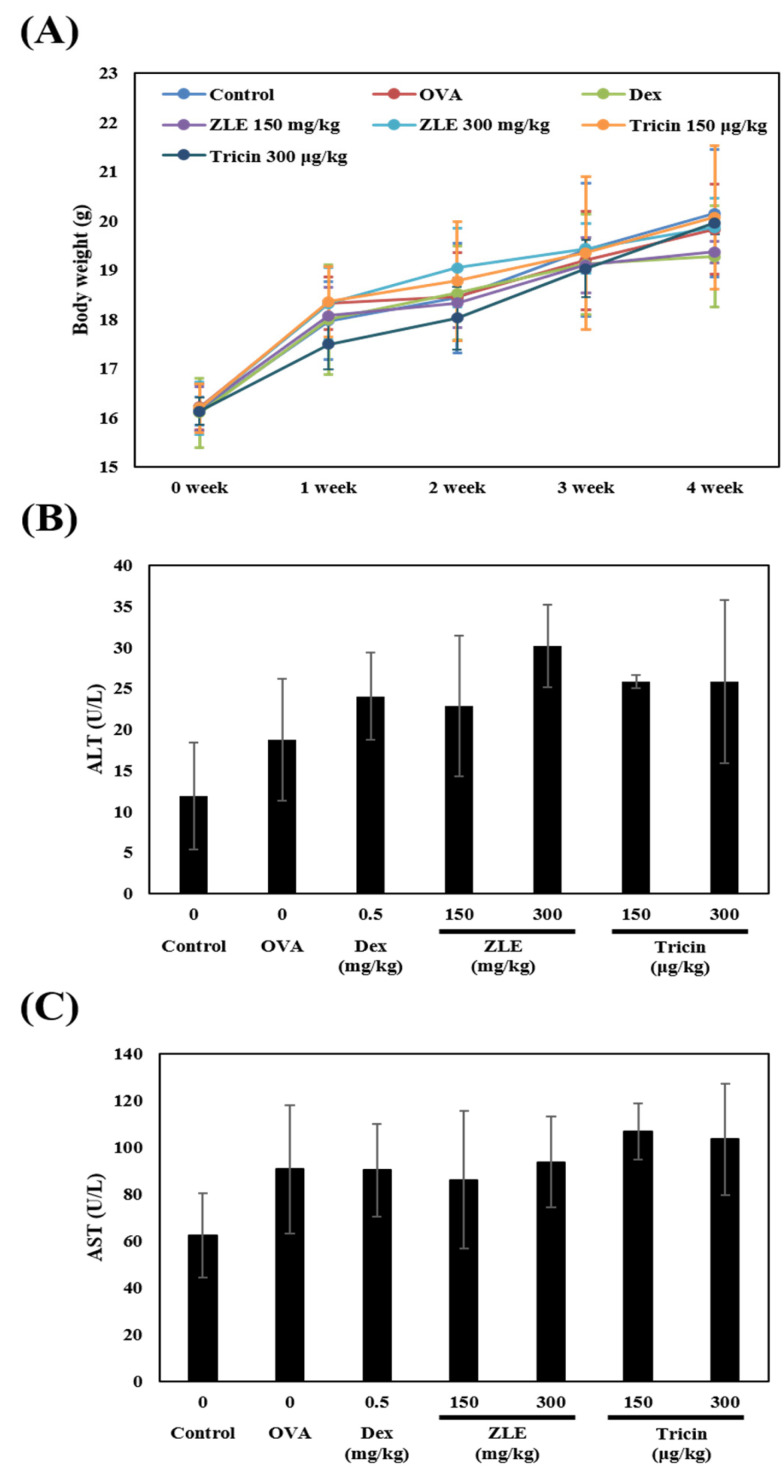
Toxicity of ZLE and tricin in mice. (**A**) Body weight, (**B**) ALT, and (**C**) AST contents in serum. Each group was calculated with Student’s *t*-test. A significant difference between each group is represented with *p* values compared to the OVA-treated group.

**Figure 3 molecules-27-03978-f003:**
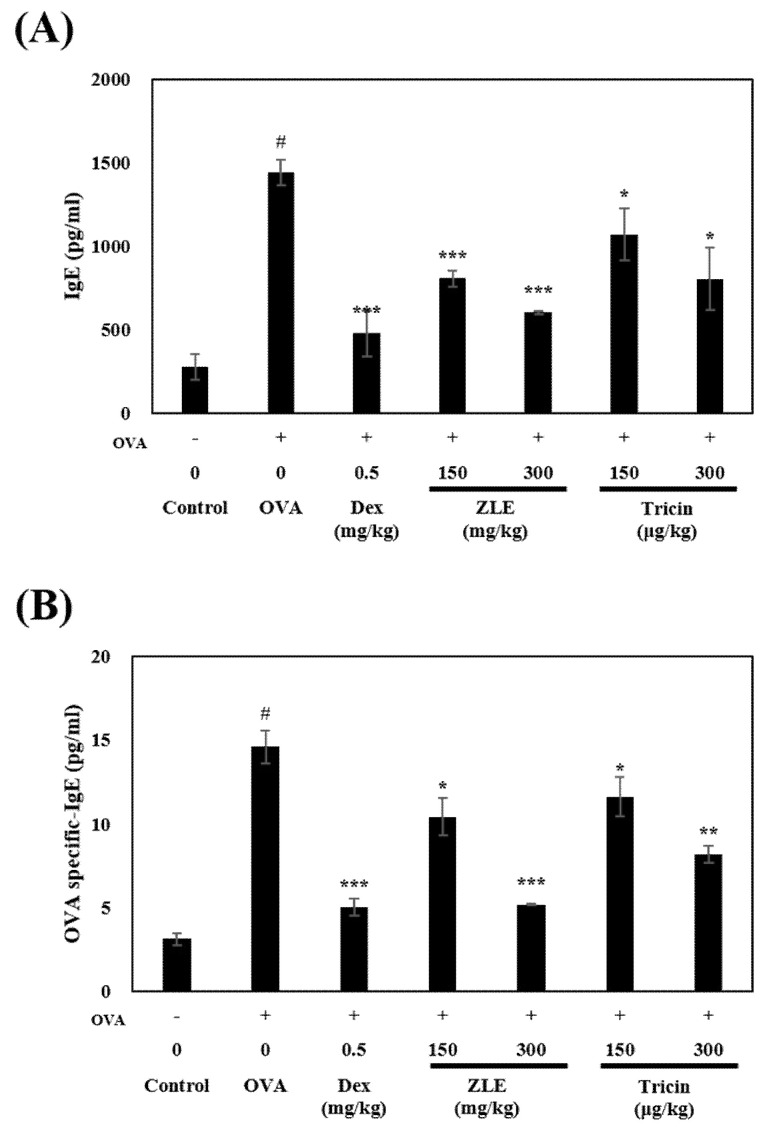
Effects of ZLE and tricin on the serum levels of IgE (**A**) and OVA-specific IgE (**B**) in OVA-sensitized mice. Groups (*n* = 5) were immunized intraperitoneally with OVA 20 μg and 2.5 mg of aluminum hydroxide on days 7 and 21, respectively. The tissues were collected 11 days after the last sensitization. The total in-serum IgE and OVA-specific IgE were measured by indirect ELISA. Each group was calculated with Student’s *t*-test. Significant differences between each group are represented with *p* values (* *p* < 0.05, ** *p* < 0.01, and *** *p* < 0.001) compared to the OVA-treated group. # *p* < 0.001 compared to the control.

**Figure 4 molecules-27-03978-f004:**
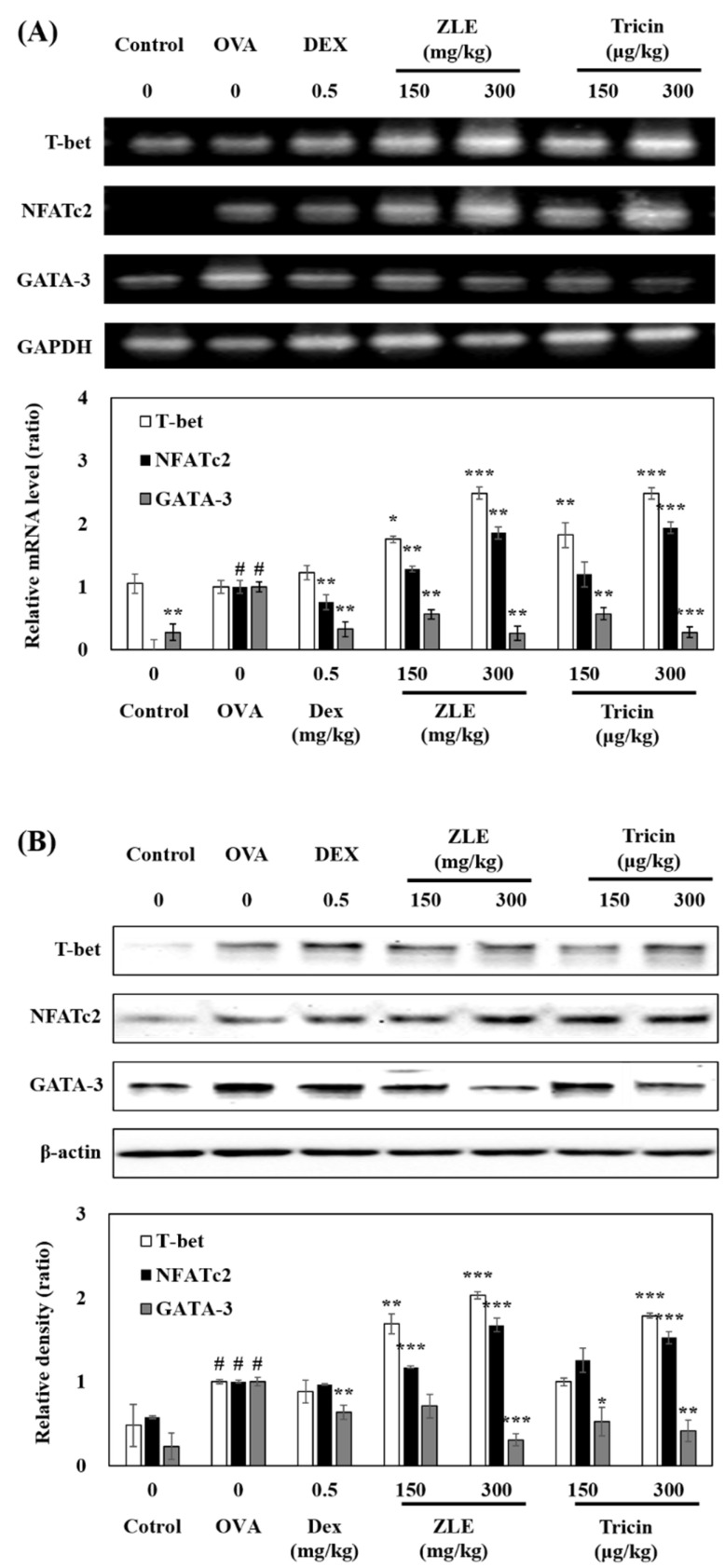
ZLE and tricin regulated the expression of T-bet, NFATc2, and GATA-3. Quantitative mRNA PCR analyses (**A**) and Western blotting (**B**). The data are presented as means ± SD values of three experiments. Each group was calculated with Student’s *t*-test. Significant differences between each group are represented with *p* values (* *p* < 0.05, ** *p* < 0.01, and *** *p* < 0.001) compared to the OVA-treated group. # *p* < 0.001 compared to the control.

**Figure 5 molecules-27-03978-f005:**
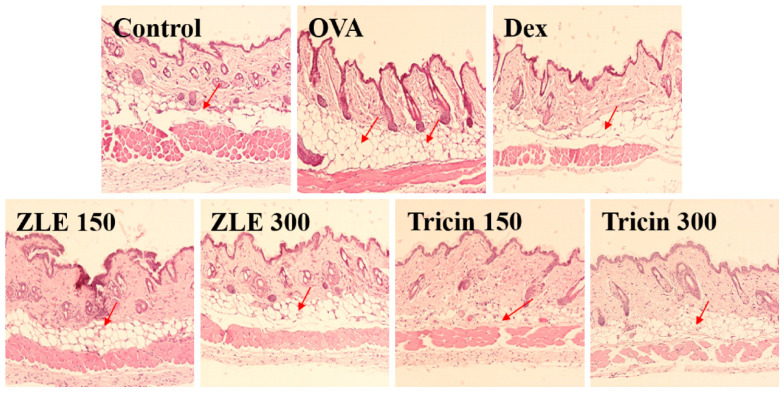
Histological changes in the skin after the oral administration of ZLE and tricin in an OVA-induced mouse model. BALB/c mouse skins obtained from the control, OVA, ZLE, and tricin groups was stained using hematoxylin and eosin (H&E). The arrowheads indicate dermal and skin thickening.

**Figure 6 molecules-27-03978-f006:**
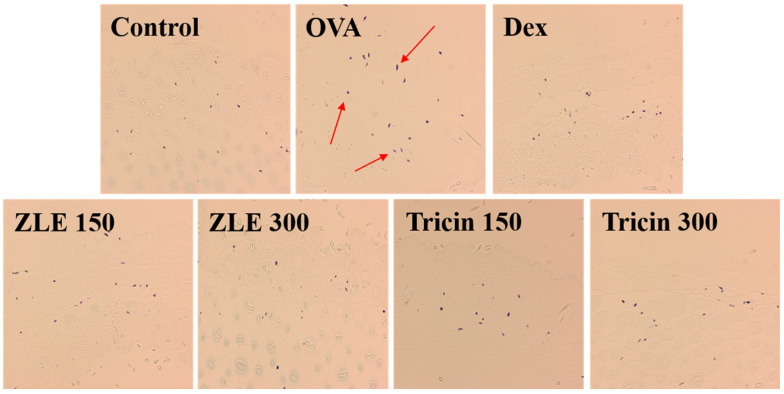
Cutaneous cell infiltration in the skin after the oral administration of ZLE and tricin in an OVA-induced mouse model. BALB/c mouse skin obtained from the control, OVA, ZLE, and tricin groups was stained using toluidine blue. The arrowheads indicate eosinophils and mast cells.

**Figure 7 molecules-27-03978-f007:**
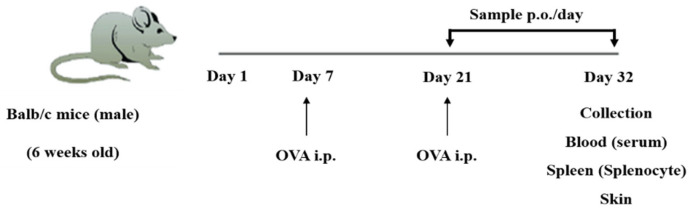
In vivo design of ZLE and tricin in OVA-treated mice. Each mouse was individually treated with OVA (20 µg, i.p.) and aluminum hydroxide (2.25 mg) on days 7 and 21, respectively. The OVA-sensitized mice (*n* = 5) were orally administrated a sample on each day from days 21 to 32. On day 32, all of the mice groups were sacrificed, and tissue samples were collected.

**Table 1 molecules-27-03978-t001:** In vivo effects of ZLE or tricin on Th1 and Th2 cytokine production by splenocytes.

Group		IL-12 (pg/mL)	IFN-γ (pg/mL)	IL-4 (pg/mL)	IL-10 (pg/mL)	IL-13 (pg/mL)	IL-5 (pg/mL)
Control	0	543.73 ± 36.99	N.D.	4.71 ± 5.07	120.38 ± 14.77	15.88 ± 1.50	10.85 ± 6.42
OVA	0	187.70 ± 3.63 #	1560.11 ± 143.52 #	640.42 ± 80.32 #	3720.26 ± 296.42 #	1586.00 ± 197.6 #	713.00 ± 118.49 #
OVA + DEX (mg/kg)	0.5	169.89 ± 25.91	1652.45 ± 507.59	139.11 ± 98.06 ***	823.85 ± 148.57 ***	788.48 ± 157.90 *	171.54 ± 20.21 **
OVA + ZLE (mg/kg)	150	315.46 ± 20.88 **	2430.53 ± 655.41 *	309.74 ± 120.15 **	2433.72 ± 201.71 **	835.90 ± 196.10 *	418.41 ± 117.44
	300	448.03 ± 17.60 ***	2882.30 ± 458.71 ***	151.84 ± 93.81 ***	1863.33 ± 383.93 **	695.52 ± 104.00 **	404.81 ± 54.71 *
OVA + Tricin (μg/kg)	150	216.39 ± 26.74	2355.13 ± 692.29	424.05 ± 182.23	2801.79 ± 175.45 *	1473.81 ± 102.80	408.57 ± 40.88 *
	300	296.39 ± 35.85 *	2697.81 ± 489.56 **	342.32 ± 87.40 **	2261.92 ± 137.54 **	1009.14 ± 87.80 *	349.61 ± 57.94 *

Each group was calculated with Student’s*t*-test. Significant differences between each group are represented with *p* values (* *p* < 0.05, ** *p* < 0.01, and *** *p* < 0.001) compared to the OVA-treated group. # *p* < 0.001 compared to the control.

**Table 2 molecules-27-03978-t002:** Experimental design.

Group		Mice	Treatment	Dose (p.o.)	Stimulate
Control	0	5	Distilled water	0	-
OVA	0	5	Distilled water	0	OVA i.p. 7, 21 days
OVA + DEX	0.5	5	Dexamethasone	0.5 mg/kg	
OVA + ZLE	150	5	ZLE	150 mg/kg	
	300	5		300 mg/kg	
OVA + Tricin	150	5	Tricin	150 μg/kg	
	300	5		300 μg/kg	

**Table 3 molecules-27-03978-t003:** Primer sequence of GATA-3, T-bet, and NFATc2 gene.

Primer		Sequence	Size (bp)
T-bet	Forward	5′-TGT GGA TGT GGT CTT GGT GG-3′	436
	Reverse	5′-ATA AGC GGT TCC CTG GCA T-3′	
NFATc2	Forward	5′-GCA CAT AAG GCC ATC AGC TCA-3′	508
	Reverse	5′-TCG CCA GAG AGA CTG GCA A-3′	
GATA-3	Forward	5′-CCT CGG CCA TTC GTA CAT G-3′	619
	Reverse	5′-CGT AGT AGG ACG GGA CGT GG-3′	
GAPDH	Forward	5′-GTT GTC TCC GAC TTC A-3′	99
	Reverse	5′-GCC CCT CCT GTT ATT ATG G-3′	

## Data Availability

Not applicable.

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
