# Peer review of "Zizania latifolia and Its Major Compound Tricin Regulate Immune Responses in OVA-Treated Mice"

_molecules, 2022, doi:10.3390/molecules27133978_

Round 1
Reviewer 1 Report
This manuscript reports the anti-allergic activity of the extract of the plant Zizania latifolia and a flavonoid tricin in mice model. Zizania latifolia extract (ZLE) and tricin were orally administrated to mice, and IgE and OVA-specific production in plasma of tricin and ZLE treated mice were investigated. Th1 and Th2 cytokines and transcription factors released in splenocytes in OVA-sensitized mice were carefully measured. It was found that tricin and ZLE could reduce plasma IgE and OVA specific-IgE levels when compared to the OVA group in the OVA-immunized mice. Moreover, tricin and ZLE were able to promote the release of Th1 cytokines IL-12 and IFN-γ, and thus inhibiting the release of Th2 cytokines (IL-4, IL-10, IL-13, and IL-5) in OVA-sensitized mice. The effects of ZLE or tricin on the expression of T-bet, NFATc2, and GATA-3 were confirmed by PCR and Western blot analysis. This work suggested that tricin and ZLE could be used to prevent allergy-related diseases through immunomodulation. Experiments in this work were well carried out. In general, this manuscript is well written, and it is highly recommended publication after minor revision. In order to improve this manuscript, please consider the comments and suggestions, which are listed below.
1. Keywords should include the words “Flavonoids”; “Flavone”; “Immunomodulation”.
2. Is the flavonoid tricin delivered to mice by intraperitoneal administration (i.p.)? Or it was delivered by oral administration? Please write clearly in the method. I found this in the result “In the present study, the oral administration of tricin and ZLE in OVA-treated mice…”, please indicate clearly in the method.
Author Response
1. Keywords should include the words “Flavonoids”; “Flavone”; “Immunomodulation”.
Response: We added the “Keywords” to the manuscript. (Page 1)
2. Is the flavonoid tricin delivered to mice by intraperitoneal administration (i.p.)? Or it was delivered by oral administration? Please write clearly in the method. I found this in the result “In the present study, the oral administration of tricin and ZLE in OVA-treated mice…”, please indicate clearly in the method.
Response: The flavonoid tricin was administered orally. We revised the “Materials and Methods” section in the manuscript. “Then, the OVA-sensitized mice (n = 5) received orally administered ZLE (150, 300 mg/kg), tricin (150, 300 µg/kg), or dexamethasone (DEX; 0.5 mg/kg) on each day.” (Page 9, line 228)

Reviewer 2 Report
This paper investigated IgE and OVA-specific production in plasma of tricin and ZLE in OVA-treated mice. The investigation confirmed the immunomodulatory effects of ZLE and tricin. I think the manuscript is well written and should draw a lot of interest from the readers of this journal. I recommend publishing after some minor revisions:
1. Page 8, table 2, why the tricin dose is 150 μmg/kg? It should be 150 mg/kg, right? Please confirm this in the revision.
2. It was mentioned the aerial parts of Z. latifolia were purchased from the Pureunsan Agricultural Association Corporation (Dongdaemun-gu, Seoul, Korea). Where were these plants harvested? We know the origin (location) of these plants could affect the amount of tricin and other compounds from these dried plants. It would be nice to see the investigation of the immunomodulatory effects of ZLE harvested from different origins.
Author Response
1. Page 8, table 2, why the tricin dose is 150 μmg/kg? It should be 150 mg/kg, right? Please confirm this in the revision.
Response: The tricin dosing level was 150 μg/kg. We revised the “Materials and Methods” section in the manuscript. (Page 8, table 2)
2. It was mentioned the aerial parts of Z. latifolia were purchased from the Pureunsan Agricultural Association Corporation (Dongdaemun-gu, Seoul, Korea). Where were these plants harvested? We know the origin (location) of these plants could affect the amount of tricin and other compounds from these dried plants. It would be nice to see the investigation of the immunomodulatory effects of ZLE harvested from different origins.
Response: We revised the “Materials and Methods” section in the manuscript. “Z. latifolia cultivated in Yeongcheon-si (Gyeongsangbuk-do, Korea) and Z. latifolia were purchased from the Pureunsan Agricultural Association Corporation (Dongdaemun-gu, Seoul, Korea).” (Page 8, line 211)

Reviewer 3 Report
The manuscript entitled “Zizania latifolia and Its Major Compound Tricin Regulate Immune Responses in OVA-treated Mice” investigated the effect of the ethanolic extract from leaves of the Manchurian wild rice Z. latifolia on Th1 and Th2 cytokines and transcription factors released in splenocytes of OVA-treated Mice.
Although the results are interesting, this study has some flaws and lacks some points which should not be published by Molecules. Please check the following reasons:
1. This study had a high probability of plagiarism since the similarity index recorded by iThenticate was 46% (without references). I strongly disagree with this ethical issue.
2. The authors have not clarified the source of tricin tested in this study. Also, the quantity of tricin in the plant and its contribution to the total activity have not been investigated.
3. The same biological activities (e.g., allergy preventative effect) of tricin has been reported and proved by many studies previously.
In summary, findings of the present study are inadequate to be published by the Molecules journal.
Author Response
1. This study had a high probability of plagiarism since the similarity index recorded by iThentica1te was 46% (without references). I strongly disagree with this ethical issue.
Response: We checked the similarity index using the iThenticalte program, and the manuscript was modified to reduce the similarity index using MDPI’s English Pre-Editing service. If necessary, we can provide an English proofreading and plagiarism-proofing warranty from MDPI.
2. The authors have not clarified the source of tricin tested in this study. Also, the quantity of tricin in the plant and its contribution to the total activity have not been investigated.
Response: We revised the “Materials and Methods” and “Introduction” part in the manuscript.
“Tricin (4′,5,7-trihydroxy-3′,5′-dimethoxyflavone) was purchased from ChromaDex. Inc. (Los Angeles, CA, USA).” (Page 8, line 206)
“Lee et al. reported that the tricin derivatives prevented the release of β-hexosaminidase in an anti-IgE-stimulated allergic reaction in RBL-2H3 mast cells, and the tricin content in Z. latifolia extract was calculated by means of LC-MS/MS analysis (0.1% tricin in Z. latifolia extract).” (Lee et al., 2015,. “Tricin derivatives as anti-inflammatory and anti-allergic constituents from the aerial part of Zizania latifolia.” Bioscience, biotechnology, and biochemistry) (Page 2, line 59)
“In our previous report, the content of tricin in the Z. latifolia extract was quantitatively analyzed and its contribution to the antiallergy effect of the plant extract was evaluated in vitro”(Lee et al., 2020, “Tricin isolated from enzyme-treated Zizania latifolia extract inhibits IgE-mediated allergic reactions in RBL-2H3 cells by targeting the Lyn/Syk pathway”, Molecules). “In the present in vitro data, we calculated that the antiallergic activity (IL-4) of tricin was approximately 55.60% compared to the Zizania latifolia extract, suggesting that the tricin in the plant strongly contributes to the antiallergic activity. In the in vivo data, the antiallergic activity (IL-4) of tricin was approximately 61.01% compared to the Zizania latifolia extract, suggesting that the tricin showed similar activity to the in vitro data.” (Page 5, line 136) Therefore, we added this to the manuscript.
3. The same biological activities (e.g., allergy preventative effect) of tricin has been reported and proved by many studies previously.
Response: We totally agreed with your opinion, and our research team searched for papers on the antiallergic effect of tricin on the Pubmed website. A few published papers on the antiallergic effect of tricin were found. Kuwabara et al. (2003) reported the antiallergic effect using mast cells (Kuwabara et al., 2003, “Tricin from a malagasy connaraceous plant with potent antihistamineic activity” Journal of Natural Products). Lee et al. (2015) confirmed the antiallergic effect of tricin and tricin derivatives isolated from Zizania latifolia (Lee et al., 2015, “Tricin derivatives as anti-inflammatory and antiallergic constituents from the aerial part of Zizania latifolia” Bioscience, Biotechnology, and Biochemistry). In addition, Lee et al. (2020) examined the antiallergic effect mechanism of tricin in RBL-2H3 mast cells (Lee et al., 2020, “Tricin isolated from enzyme-treated Zizania latifolia extract inhibits IgE-mediated allergic reactions in RBL-2H3 cells by targeting the Lyn/Syk pathway”, Molecules). These reports showed that tricin effectively inhibited allergy-related biomarkers, such as cytokines (e.g., IL-4 and TNF-α) or plastagladins, but these data were simply a case of comparing the antiallergic effect in an in vitro system. Yin et al. (2003) revealed the inhibitory effect of inflammatory asthma using LPS-OVA-treated mice. However, the immunomodulatory effects of trocin on Th1/Th2 cytokines were not confirmed in this study. (Yin et al., 2017, “Tricin's inhibitory effects on TLR4/MyD88/NF-κB pathway of alveolar macrophages in asthma mice” Chinese Traditional Patent Medicine). In vitro studies in 2022 showed that tricin has excellent antiallergic effects, but animal studies were not performed. Therefore, the research team confirmed that the antiallergic effect of tricin in vivo is a new research topic.

Reviewer 4 Report
Manuscript described ant allergic effect of Zizania latifola ethanolic extract and tricin (the flavonoid component of the extract). Experimental part is well planned and the results are clearly described. However, I have some questions/suggestions for Authors.
1) On what basis the dose of tricin was chosen? Does it correspond to the content in the extract? This would help to assess whether tricin is the main compound responsible for the extract activity. Moreover, it would be useful to determine the amount of tricin in the extract. If it is not possible, estimate the quantity based on the literature.
2) Lines 67 – 73 should not be placed in Introduction because they summarize the study. Introduction should only give relevant background and describes the aim of the study
3) Figures should be placed below first mention in the text (see Fig. 2)
4) Figure 2: Legend for Figure should be more informative
5) Line 80: “The body weight changes in ZLE or tricin administration did not change in any group” – unclear sentence
6) Line 84: Add the reference range
7) Table 1: Control IL13: 15.88 ± 150 – probably lack of dot
8) Line 211: Was it Soxhlet extraction or heat reflux extraction? Ratio of dried material to solvent should be added.
9) Lack of information on tricin in Material and Method. Was it isolated from plant material?
10) Move conclusions in a separate section
Minor comments:
- line 61: „aerial portion” – replace by „aerial part”; „They reported (…)” – replace by name of the author
- lack of italic (e.g. Fig.2 legend, line 80, 138..)
Author Response
1) On what basis the dose of tricin was chosen? Does it correspond to the content in the extract? This would help to assess whether tricin is the main compound responsible for the extract activity. Moreover, it would be useful to determine the amount of tricin in the extract. If it is not possible, estimate the quantity based on the literature.
Response: We revised the “Introduction” and “Results and Discussions” sections in the manuscript.
Lee et al. reported that the tricin derivatives prevented the release of β-hexosaminidase in an anti-IgE-stimulated allergic reaction in RBL-2H3 mast cells, and the tricin content in Z. latifolia extract was calculated by means of LC-MS/MS analysis (0.1% tricin in Z. latifolia extract). (Lee et al., 2015,. “Tricin derivatives as anti-inflammatory and anti-allergic constituents from the aerial part of Zizania latifolia.” Bioscience, biotechnology, and biochemistry) (Page 2, line 59)
In this study, 0.1% tricin was used for each ZLE concentration to confirm the efficacy of tricin on ZLE. In our previous report, the content of tricin in the ZLE was quantitatively analyzed and its contribution to the anti-allergy effect of the plant extract was evaluated in vitro. (Lee et al., 2020, “Tricin isolated from enzyme-treated Zizania latifolia extract inhibits IgE-mediated allergic reactions in RBL-2H3 cells by targeting the Lyn/Syk pathway”, Molecules). (Page 5, line 137)
2) Lines 67 – 73 should not be placed in Introduction because they summarize the study. Introduction should only give relevant background and describes the aim of the study
Response: We removed lines 67-73 (Page 2).
3) Figures should be placed below first mention in the text (see Fig. 2)
Response: We moved Figure 2 to the “Materials and Methods” and changed it to Figure 7 (Page 9).
4) Figure 2: Legend for Figure should be more informative
Response: We added the information in Figure 7. (Changed from Figure 2 to Figure 7.) “Figure 7. In vivo design of ZLE and tricin in OVA-treated mice. Each mouse was individually treated with OVA (20 µg, i.p.) and aluminum hydroxide (2.25 mg) on days 7 and 21, respectively. The OVA-sensitized mice (n = 5) were orally administrated a sample on each day from days 21 to 32. On day 32, all of the mice groups were sacrificed, and tissue samples were collected.” (Page 9, line 242)
5) Line 80: “The body weight changes in ZLE or tricin administration did not change in any group” – unclear sentence
Response: The manuscript was revised. “No changes were observed for body weight, regardless of whether ZLE or tricin was administered” (Page 2, line 72)
6) Line 84: Add the reference range
Response: We added the references to the manuscript (Reference 17-20). (Page 2, line 76)
- Lee, M. Y.; Lee, N. H.; Jung, D. Y.; Lee, J. A.; Seo, C. S.; Lee, H. Y.; Kim, J. H.; Shin, H. K. Protective effects of allantoin against ovalbumin (OVA)-induced lung inflammation in a murine model of asthma. Int. Immunopharmacol.2010, 10, 474-480. https://doi.org/10.1016/j.intimp.2010.01.008.
- Lim, H. B.; Kim, S. H. Inhallation of e-cigarette cartridge solution aggravates allergen-induced airway inflammation and hyper-responsiveness in mice. Toxicol. Res.2014, 30, 13-18. https://doi.org/10.5487/TR.2014.30.1.013
- Lee, M.Y.; Seo, C.S.; Ha, H.K.; Jung, D.Y.; Lee, H.Y.; Lee, N.H.; Lee, J.A.; Kim, J.H. Lee, Y.K. Son, J.K.; Shin, H. K. Protective effects of Ulmus davidiana var. japonica against OVA-induced murine asthma model via upregulation of heme oxygenase-1. J. Ethnopharmacol.2010, 130, 61-69. https://doi.org/10.1016/j.jep.2010.04.011
7) Table 1: Control IL13: 15.88 ± 150 – probably lack of dot
Response: The control IL-13 was 15.88 ± 1.50. We revised “Table 1” in the manuscript (Page 5).
8) Line 211: Was it Soxhlet extraction or heat reflux extraction? Ratio of dried material to solvent should be added.
Response: We revised the manuscript. “Dried leaves of Z. latifolia underwent extraction with 10 times its weight in 70% ethanol (Duksan Science, Seoul, Korea) at 80°C for 6 h. The extracted solution was concentrated and lyophilized to produce ZLE. The lyophilized ZLE confirmed to have a yield of 15%.” (Page 8, line 213)
9) Lack of information on tricin in Materials and Methods. Was it isolated from plant material?
Response: We revised the “Materials and Methods” section in the manuscript. “Tricin (4′,5,7-trihydroxy-3′,5′-dimethoxyflavone) was purchased from ChromaDex. Inc. (Los Angeles, CA, USA).” (Page 8, line 206)
10) Move conclusions in a separate section
Response: We moved “3. Conclusions” in the manuscript.
Minor comments:
- line 61: „aerial portion” – replace by „aerial part”; „They reported (…)” – replace by name of the author
Response: We added the name of the author. (Page 1, lines 42, page 2, lines 56, 61, page 8, line 211, page 9, line 248, page 10, line 282)
- lack of italic (e.g. Fig.2 legend, line 80, 138..)
Response: We revised the manuscript. (Page 9, Fig. 7 legend [Line 242, Changed from Figure 2 to Figure 7.], page 5, line 132, page 9, line 226, page 11, lines 320, 326, 329, 333, 334, page 12, lines 364, 365)

Round 2
Reviewer 3 Report
The authors addressed most of the issues that I pointed out.
However, authors should improve the discussion part by focusing on tricin in terms of quantity in natural materials, structure-related biological activities, and the potential sources of this molecule.
I agree that authors have already highlighted the novelty in the pharmaceutical property of tricin, however, to be more convincible and appropriate to Molecules journal, the authors should consider this point. For example, you can compare the content of tricin in this wild rice (Z. latifolia) with that of rice variety (Oryza sativa). You can refer to this article 10.3390/foods8120602.
I suggest an acceptance after minor revisions based on the recommendations above.
Author Response
Response: Thank you for your comments. As per your suggestions, we have added references and material relating to the biofunctional properties, structure, and contents of tricin in the manuscript as follows:
“The reports of Mohanlal et al. [16], Kwon and Kim [17], and Quan et al. [18] calculated the tricin content of Oryza sativa at 48.6 mg/kg, 32.9 mg/kg, and 0.55 mg/kg, respectively.” (Page 2, line 63)
“Indeed, Jiang et al. [20] revealed that tricin has effectively the biological properties by functional groups such as phenolic hydroxyl groups, carbonyl groups, methoxyl groups.” (Page 2, line 67)
